# Effective Treatment of Neurological Symptoms with Normal Doses of Botulinum Neurotoxin in Wilson’s Disease: Six Cases and Literature Review

**DOI:** 10.3390/toxins13040241

**Published:** 2021-03-29

**Authors:** Harald Hefter, Sara Samadzadeh

**Affiliations:** Department of Neurology, University Hospital of Düsseldorf, Moorenstrasse 5, D-40225 Düsseldorf, Germany; Harald.Hefter@med.uni-duesseldorf.de

**Keywords:** Wilson’s disease, neurological symptoms, botulinum neurotoxin type A, dose adjustment, reduced compliance

## Abstract

Recent cell-based and animal experiments have demonstrated an effective reduction in botulinum neurotoxin A (BoNT/A) by copper. Aim: We aimed to analyze whether the successful symptomatic BoNT/A treatment of patients with Wilson’s disease (WD) corresponds with unusually high doses per session. Among the 156 WD patients regularly seen at the outpatient department of the university hospital in Düsseldorf (Germany), only 6 patients had been treated with BoNT/A during the past 5 years. The laboratory findings, indications for BoNT treatment, preparations, and doses per session were extracted retrospectively from the charts. These parameters were compared with those of 13 other patients described in the literature. BoNT/A injection therapy is a rare (<4%) symptomatic treatment in WD, only necessary in exceptional cases, and is often applied only transiently. In those cases for which dose information was available, the dose per session and indication appear to be within usual limits. Despite the evidence that copper can interfere with the botulinum toxin in preclinical models, patients with WD do not require higher doses of the toxin than other patients with dystonia.

## 1. Introduction

Botulinum neurotoxin type A (BoNT/A) is one of the most potent biological toxins [1], may cause foodborne botulism [2,3], and continues to be a bioterrorism threat [4,5]. Therefore, effective anti-BoNT/A compounds are needed to reduce the risk of life-threatening BoNT/A overdose. The action of BoNT/A results from the binding of the heavy chain to a cell membrane, endocytosis, and translocation of the light chain (LC) into the cytosol of the cell. There, the LC zinc metalloprotease cleaves synaptosomal-associated protein 25 (SNAP-25), a component of the soluble NSF attachment protein receptor (SNARE) complex responsible for docking vesicles at the presynaptic membrane [6,7,8]. Therefore, the majority of anti-BoNT/A compounds target LC because of their key role in exocytosis [9]. However, as long as such substances act only extracellularly as antibodies, for example, the enzymatic activity of the LC cannot be stopped. This seems to be different for metal complexes, providing rationale for the development of a novel class of LC inhibitors. Copper and mercury are highly significant LC inhibitors. A key binding interaction between copper and Cys165 in the BoNT/A LC has been analyzed and described. Extracellularly applied ligand–copper complexes at low concentrations effectively reduce intracellular LC cleavage of SNAP-25. Administration of copper complexes in life-threatening BoNT/A-treated rodents effectively delayed BoNT/A mediated lethality [9].

Both cell-based and animal experiments suggest that the efficacy of botulinum toxin treatment may be reduced in patients with Wilson’s disease, who suffer from deficient copper transport and elevated copper storage. In Wilson’s disease, due to a recessively inherited deficit on chromosome 13, the biliary copper excretion from the liver into the stool is disturbed, leading to a copper overload primarily of the liver and secondarily of the entire organism [10]. After hepatic manifestation, neurological symptoms are the most frequent clinical symptoms of Wilson’s disease (WD). Initial neurological presentation occurs in 18–68% of patients, with mean age at symptom onset of 20–30 years [11,12]. The spectrum of neurological symptoms in WD is broad [10,12,13]. 

Standard treatment of WD is oral therapy with chelating substances that enhance urinary copper excretion. Unfortunately, not all neurological symptoms respond to this therapy equally well [14]. Tremor, which is observed in up to 55% of patients with neurological WD, responds excellently; dystonia, which also frequently occurs in neurological WD, may persist or even progress under chelating therapy in up to 65% of patients. Parkinsonian symptoms such as bradykinesia, rigidity, hypomimia, gait and posture disturbances, dysarthria, dysphagia, and drooling may occur in neurological WD [13,14,15,16], which require symptomatic treatment. The use of L-dopa, trihexyphenidyl, benzodiazepine, and botulinum neurotoxin has been suggested for symptomatic treatment in WD [17,18,19,20]. However, reports on the symptomatic treatment in WD with botulinum neurotoxin are rare. One possibility might be that BoNT/A has only a minimal effect in WD. This is supported by the above-mentioned theoretical paper demonstrating that copper reduces the activity of BoNT/A so effectively that there are various attempts to develop copper complexes as a treatment for botulism [9].

With the present series of six patients, which is the largest one to date for the symptomatic treatment with botulinum toxin of a broad spectrum of symptoms in WD, we summarize our experience with a special focus on the doses and efficacy of BoNT/A in WD.

## 2. Report of Patients

### 2.1. WD and Palmar Hyperhidrosis (Patient 1)

This 30-year-old woman was diagnosed with WD at the age of 10 years. She was asymptomatic when treatment was initiated. Due to a severe skin reaction, she was switched from a low dose (600 mg) of D-penicillamine (DPA) to 1200 mg Trientine^®^. After puberty, she developed progressive general hyperhidrosis and extreme speech hastening, which is her main symptom. She did not accept therapy with a higher dose of Trientine^®^. She has normal intelligence but did not finish her education during the last ten years.

She receives an injection with 100 U of onabotulinumtoxin A (onaBoNT/A; Botox^®^) per hand because of palmar hyperhidrosis once or twice a year. She reports a good (about 60% improvement during the first 3 months after the injection) response to BoNT/A; the effect lasts for months. 

### 2.2. WD and Segmental Dystonia (Patient 2)

This 66-year-old woman and her younger sister were diagnosed with WD at the ages of 12 and 10 years, respectively, when a family screening was performed. A metabolic disorder was suspected because both sisters had elevated liver enzymes and she had developed tremor and impaired handwriting.

About 6 years later because of hip dysplasia, she had to be operated on and a hip replacement was performed. After the operation, a prolonged wake-up phase was noticed. During the next years, she developed complex segmental dystonia involving the neck and upper trunk muscles. However, she refused to increase the dose of DPA beyond 900 mg. 

She was treated with 500 to 1000 U of abobotulinumtoxin A (aboBoNT/A; Dysport^®^) and experienced moderate improvement, but was satisfied with this treatment. Under treatment with 900 mg of DPA and about 750 U of aboBoNT/A in the meantime, her situation was stable over years. She was a smoker and had chronic coughs.

When she complained of general weakness and intensive coughs, lung cancer was detected, but she refused to undergo chemotherapy or surgical intervention. She died within 7 months (Figure 1).

### 2.3. WD and Generalized Dystonia and Hypersalivation (Patient 3)

This 64-year-old woman was diagnosed with WD when she was 15 years old. In 1988, she presented the first time in our department with generalized dystonia. Dose was increased from 900 mg of DPA up to 2700 mg without any relevant change in the dystonia. During the next years, dysarthria became so severe that she had to use a computer-assistive device. Swallowing became difficult and she refused to be switched to Trientine^®^.

With progressive dysarthria and difficulties in swallowing, she also developed severe hypersalivation, which was treated from time to time with 200 U of incobotulinumtoxin A (incoBoNT/A; Xeomin^®^). During the first 3 months after the injection, she had a good response. Thereafter, the effect slowly declined, but she did not want to attend the botulinum toxin outpatient clinic more frequently (Figure 2).

### 2.4. WD and Generalized Dystonia (Patient 4)

The history of this patient has already been mentioned in two articles [21,22]. In short, he was diagnosed at the age of 18 years and treated with 600–900 mg of DPA. Hyperkinesia and long tract involvement improved under medication, but dysarthria progressed. During the following years, the patient moved to another city, studied computer sciences, and became an information technology (IT) specialist. He did not increase the medication although he became so dysarthric that only his wife was able to understand and translate him. When he returned to a nearby city and presented again in our institution, he had developed severe generalized dystonia.

Due to a severe pain syndrome with pain in the lower back more on the left than on the right side and in the left arm, he had (as an IT specialist) looked for help on the Internet and therefore presented to be treated with BoNT/A, not for control of WD.

He is highly cooperative concerning his BoNT/A injections but not WD. He is injected every 3 months with 200 U of incoBoNT/A into the back and 200 U into the left arm, with a good response (Figure 3).

### 2.5. WD and Multifocal Dystonia and Hypersalivation (Patient 5) 

This 52-year-old man was diagnosed after high school when he started to study medicine. He developed writing difficulties, tremors, and fatigue, as well as a lack of concentration. Finally, he developed juvenile Parkinsonian syndrome with drooling. He was listed for liver transplantation, but liver dysfunction and neurological symptoms recovered under treatment with 900 mg of DPA. He gave up studying medicine and became a physiotherapist. The Parkinsonian syndrome improved, but multifocal dystonia persisted with dysarthria, cervical dystonia, and foot dystonia. Due to a disc prolapse, he was operated on at the cervical spine when he was 47 years old. He tended to treat his motor problems with physiotherapy and remained on the low dose of 900 mg of DPA. Over the years, he gradually worsened: his cervical dystonia and dysarthria became severe. When he was 50 years old, he experienced a left hemispheric stroke probably due to a dissection of the left carotid artery, but an open foramen oval with a paradoxical embolus might have also been the cause of the stroke. 

Before the stroke, the patient had already been treated with 500 U of aboBoNT/A or 200 U of incoBoNT/A to improve neck pain. After the stroke, the patient became anarthric and started to drool again. In addition to the treatment of the cervical dystonia, the patient was also treated because of the hypersalivation with 100 U of incoBoNT/A into the parotid glands with success (Figure 4). 

### 2.6. WD and Spasmodic Dysphonia (Patient 6)

The diagnosis of WD was made when the patient was 5 years old. Development and childhood were normal under continuous medication. When the patient was married, he left his parents and decided to stop the medication. Even after a first visit to the department of gastroenterology in Düsseldorf, he continued cessation of medication. In the following 3 years, he became dysarthric and suffered from an impulse control deficit. He was divorced and decided to live with his parents again. In 2014, he survived an embolic infarction of the lungs and decided to start with a low dose of DPA again. Due to increasing difficulties in swallowing and loss of 14 kg bodyweight, he was referred to be analyzed for the presence of malignancy, which was not confirmed. During that time, he became anarthric and used his mobile phone for communication. The consulting neurologist and psychiatrist both agreed on the diagnosis of a psychogenic speech disorder because of the severe psychosocial problems and recommended treatment with Seroquel^®^. The speech therapist insisted on the diagnosis of organic dysarthrophonia.

Five months later, the patient presented in our institution. He was completely anarthric and had a juvenile Parkinsonian syndrome, which rapidly improved after the withdrawal of Seroquel^®^. Dose of DPA was increased up to 2700 mg and dysarthria was treated with 5 U of onaBoNT/A or 10 U of aboBoNT/A per cricoarytenoideus muscle. Injections were performed from the outside. Clinical score, speech, and laboratory findings slowly but continuously improved excellently (Figure 5; right side). After 2 years, BoNT/A injections could be discontinued. Laboratory findings yielded a normal copper excretion after 3 days of no medication. Thereafter, the patient reduced and withdrew medication a second time and worsened rapidly again. Two more injections of BoNT/A became necessary and the dose was increased again to the former level. In the following months and years, the patient stabilized, did not need further BoNT/A injections, and his impulse control deficit considerably improved. Nevertheless, the patient is unable to perform regular work (Figure 5).

## 3. Cases in the Literature

Table 2 provides an overview of the cases presented in the literature to date. Unfortunately, no doses are presented, with only one exception. Teive et al. [23] reported that 35 U was injected per lateral pterygoid and 30 U in the submentalis muscle complex. Based on our experience, this is not an unusually high dose. Patients responded with a mild-to-moderate improvement, which is within the range of responses reported in the literature for this difficult-to-treat type of dystonia [24]. For the patient with hand dystonia, it is only reported that a series of injections with aboBoNT/A had been performed and that the hand recovered completely. In this case, the effect of BoNT/A cannot be distinguished from the effect of withdrawal of the dystonia-inducing medication.

## 4. Discussion

### 4.1. BoNT/A Is Effective in Normal Doses in WD

The number of WD patients including the present case series reported to be treated with botulinum neurotoxins is fairly small (*n* = 19). Unfortunately, the preparation and dose are often missing in the reports. None of the reports indicated inefficacy of BoNT therapy or response only to unusual high doses. In our case series, mild to very good efficacy was observed during the treatment of all six patients. The spectrum of indications for treatment with BoNT in the 19 WD patients was broad (Table 1 and Table 2). In those indications where no effectiveness would have been noticed immediately (dysarthria in patient 6 and hand function in Litwin et al. [26]) and for which low doses were used, efficacy was the best. In summary, the present and other cases with clinical data reported in the literature to date do not provide any hint that normal doses of BoNT are not effective in WD. The reason for why this is the case is not obvious. The mechanism of how the LC of BoNT/A and copper interact intracellularly is well-known; extracellular administration of copper complexes effectively reduces LC-mediated cleavage of SNAP-25 [9]. Thus, for the interaction between copper and the LC of BoNT/A, sufficiently high levels of copper have to be available intracellularly. As long as copper is irreversibly bound intracellularly to metallothionein [10], which is usually the case in long-term-treated WD patients, this copper accumulation does not influence BoNT action. However, the normal response to BoNT/A treatment of muscles and glands in WD, as demonstrated in the present paper, indicates that the crossly fluctuating and not always elevated levels of free copper in the serum of not optimally cooperating WD patients, as in our series, are not high enough to reduce BoNT activity to a clinically relevant extent.

### 4.2. BoNT/A Is Only Used in Severely Affected Patients 

The WD treatment strategy is to diagnose and treat it and eliminate copper as early and as possible. With this strategy, the majority of WD patients can be kept in an asymptomatic state. If neurological symptoms have become manifest, the use of higher doses is necessary to reduce the neurological symptoms [27,28]. A mild slowness or bradykinesia may persist, but in most WD patients, symptoms can be reduced to such a degree that employment is preserved [13]. Therefore, in most WD patients, there is no need for symptomatic treatment with botulinum neurotoxin or other oral medication than copper chelating drugs [17]. Therefore, it does not matter that, worldwide, WD patients are treated mainly by hepatologists with little experience with BoNT/A; however, in exceptional or incompliant patients, a clear indication for treatment with BoNT/A may exist. This is demonstrated by our case series. All these six patients were not optimally compliant (compare laboratory findings in Table 1) for different reasons. These few patients can be referred to movement disorder specialists for BoNT/A treatment.

### 4.3. In the Majority of WD Patients, There Is No Indication for BoNT/A Treatment

In the majority of WD patients, neurological symptoms respond quite well to copper elimination therapy [13,14]. Only in a small percentage of WD patients do severe neurological symptoms persist. If these patients undergo liver transplantation, neurological symptoms often respond quite well so that BoNT/A may be performed to bridge to transplantation and provide the patient some relief, as described by Demasio et al. [25]. Even when a WD patient is newly diagnosed, there is no urgent need to initiate BoNT therapy as well. However, for patient 6, who became anarthric after the withdrawal of medication, or patient 5, who suffered from severe drooling after a stroke, it may greatly benefit the patient to be treated with BoNT transiently.

## 5. Conclusions

In cell-based assays and rodent experiments, copper effectively reduces BoNT/A action [9]. This recent finding raises the question as to whether patients with Wilson’s disease suffering from neurological manifestations can be treated with normal doses of BoNT/A. Neither the present case series nor the literature present any clinical evidence that unusually high doses are needed for effective symptomatic treatment with BoNT/A in neurological WD. Patients with WD respond fairly well to BoNT/A injections. Therefore, a more widespread use of BoNT/A is recommended for symptomatic treatment in WD than the literature to date reflects.

## Figures and Tables

**Figure 1 toxins-13-00241-f001:**
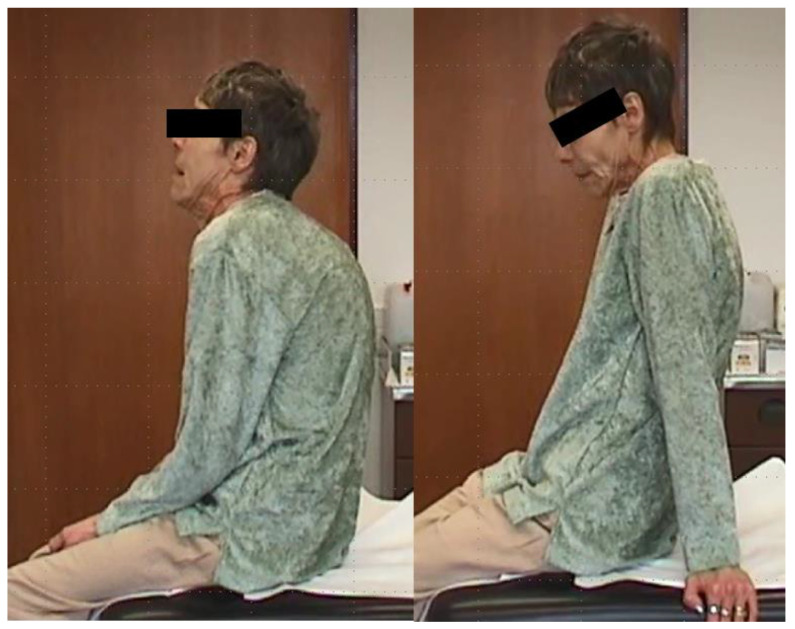
Wilson’s disease (WD) and segmental dystonia (patient 2). During upright sitting, shoulders were pulled forward, and the head was drawn back and downwards (**left side**). When she leaned back, shoulders could be moved back and the retrocaput component disappeared (**right side**).

**Figure 2 toxins-13-00241-f002:**
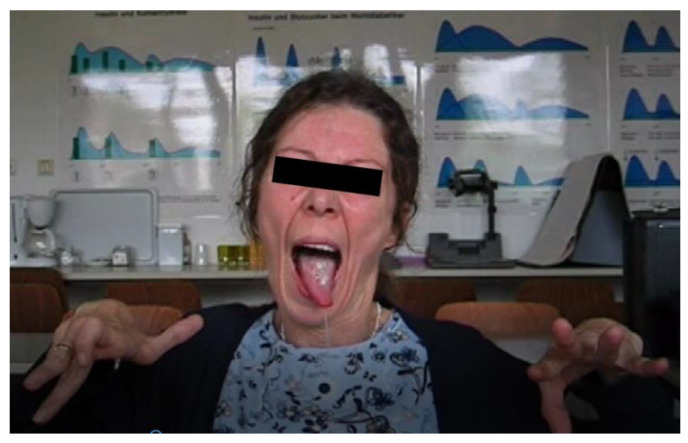
WD and generalized dystonia (Meige syndrome, oromandibular dystonia, cervical dystonia, trunk, and limb dystonia) and hypersalivation (patient 3).

**Figure 3 toxins-13-00241-f003:**
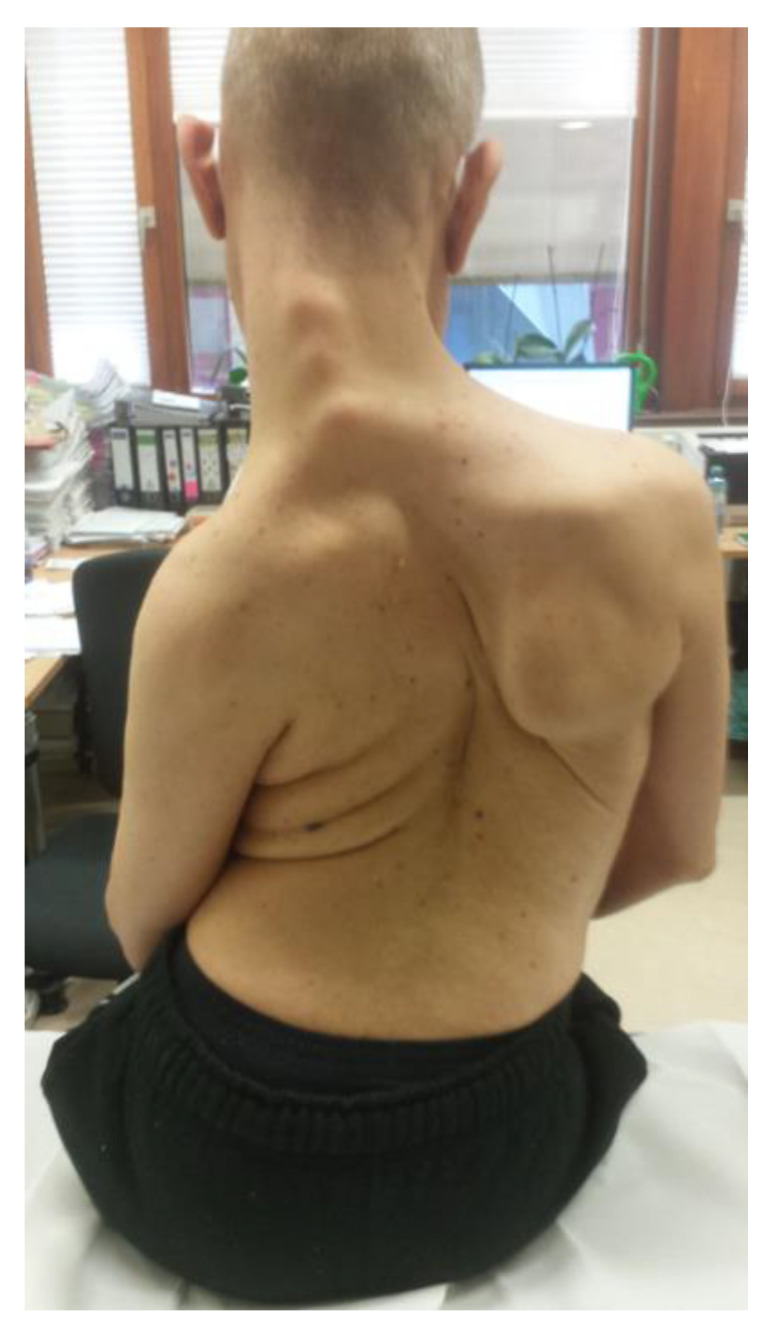
WD and generalized dystonia (patient 4).

**Figure 4 toxins-13-00241-f004:**
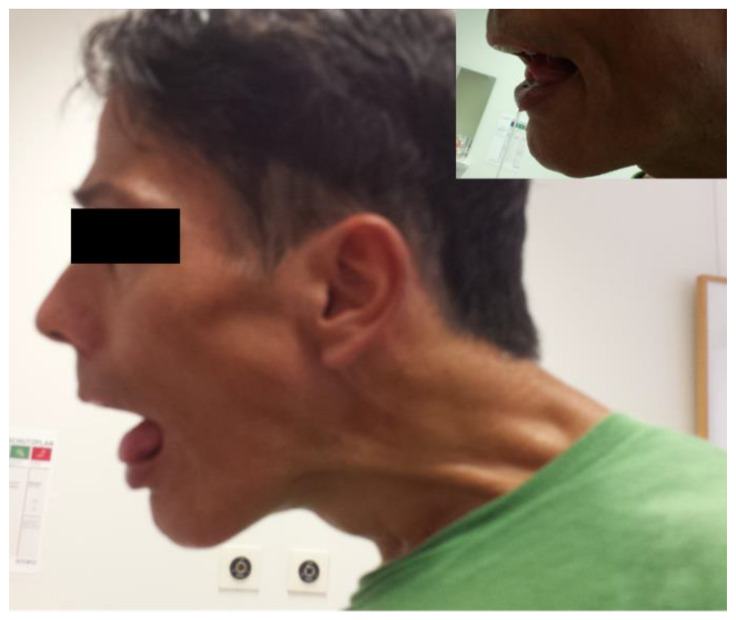
WD and multifocal dystonia and hypersalivation (see upper insert; patient 5).

**Figure 5 toxins-13-00241-f005:**
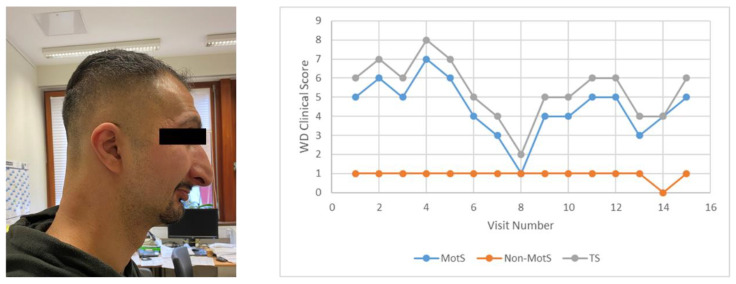
WD and spasmodic dysphonia (patient 6; **left**). (**right**) Development of clinical scores (for details, see Table 1), demonstrating excellent improvement after onset of copper chelating therapy. After a second withdrawal of the medication, the patient worsened again. He did not reach the level of improvement again that he had had before the second withdrawal of the medication.

**Table 1 toxins-13-00241-t001:** Demographical and treatment-related data of WD and botulinum neurotoxin A (BoNT/A) therapy.

Parameter	Patient 1	Patient 2	Patient 3	Patient 4	Patient 5	Patient 6
**Demographical data**						
Age	30	66	64	50	52	35
Sex	FEMALE	FEMALE	FEMALE	MALE	MALE	MALE
Age at first diagnosis (years)	10	12	15	18	18	5
WD therapy	Trientine^®^	DPA	Trientine^®^	DPA	DPA	DPA
Clinical score *						
MotS	2	5	8	9	9	5
Non-MotS	0	0	3	0	2	1
TS	2	5	11	9	11	6
**Laboratory findings **** **(of the last visit before recruitment into this study: grey windows indicate values out of the normal range)**						
Cerulo (mg/dL)	<7	<7	14	10	<7	<7
CU (mg/dL)	0.02	0.21	0.49	0.40	0.60	0.05
AST (U/L)	25	21	41	34	52	15
CHE (U/L)	6079	5194	5149	6744	4290	3870
24 h-CU mg/d	0.041	0.045	0.092	0.037	0.059	0.377
**BoNT/A Therapy** **(data of the last visit before recruitment: treatment related data and laboratory findings result from the same visit)**						
Indication	Palmar hyperhidrosis	Segmental dystonia	Hypersalivation	Generalized dystonia	Cervical dystonia hypersalivation	Spasmodic dysphonia
Preparation	ona BoNT/A	Abo BoNT/A	inco BoNT/A	inco BoNT/A	inco BoNT/A	ona- or inco BoNT/A
Dose of BoNT/A	100 U/hand	500–1000 U for neck	100 U/gland	200 U for trunk 200 U for left arm	200 U for CD 100 U/gland	5–10 U/ side
Recommended dose range	100 U/hand	500–1000 U for CD	50–100/gland	Off-label	200 U for CD 100 U/gland	5–10 U/ side
Efficacy	Good	Moderate	Moderate	Good	Mild/ Moderate	Very good
Side effects	Pain during injection	None	None	None	None	None

* Clinical score = 7 motor items (dystonia, dysarthria, bradykinesia (reduced frequency of alternating finger movements or alternating tongue movements), tremor, gait disturbance, oculomotor deficits, cerebellar abnormalities (during the finger/nose test or during the knee/heel test or during rebound testing)) as well as three non-motor abnormalities (reflex abnormalities, sensory abnormalities, and neuropsychological and psychiatric abnormalities (such as anxiety, depression, hallucinations, and cognitive impairment) are scored whether these abnormalities are absent (0) or only mildly (1), moderately (2), or severely (3) present. The motor sub-scores are summed up to yield a Motor Score (MotS: 0–21); the three non-motor sub-scores are summed up to a Non-Motor Score (N-MotS: 0–9); and the sum of MotS and N-MotS yields the Total Score (TS: 0–30) (for details see [13]). ** Cerulo, Ceruloplasmin: normal range in our clinical laboratory (NR): 20–60 mg/dL; CU, serum copper: NR: 0.7–1.5 mg/L; AST, alanine aminotransferase: NR: <35 U/L; CHE, cholinesterase: NR: 5120–12.920 U/L; 24 h-CU, 24-h urine copper excretion: NR: <0.04 mg/d.

**Table 2 toxins-13-00241-t002:** Case reports in the literature.

Parameter	Demasio et al., 2008 [25]	Litwin et al., 2013 [26]	Hölscher et al., 2010 [20]	Teive et al., 2012 [23]
N	4	1	3	5
Severity of WD	Severe dystonia, 3 patients to be transplanted	Hand dystonia induced by antidepressants	2 with dystonia, 1 with tremor	General dystonia
Indication	Pain Limb dystonia	Hand dystonia	Dystonia Tremor	Jaw opening dystonia
Preparation	n.m.	aboBoNT/A	n.m.	onaBoNT/A
Dose	n.m.	n.m.	n.m.	100 U
Efficacy	some functional recovery	complete normalization	n.m.	Mild to moderate improvement
Side effects	n.m.	n.m.	n.m.	3/5 mild dysphagia

n.m. = not mentioned.

## Data Availability

Data available on request due to restrictions eg privacy or ethical. The data presented in this study are available on request from the corresponding author.

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
