# Peer review of "Effective Treatment of Neurological Symptoms with Normal Doses of Botulinum Neurotoxin in Wilson’s Disease: Six Cases and Literature Review"

_toxins, 2021, doi:10.3390/toxins13040241_

Round 1

Reviewer 1 Report

The manuscript describes 6 Wilson’s disease patients that are symptomatically treated with Botulinum injections. The Authors use experimental results as a preface to the study, however, this manuscript is not able to test this hypothesis. The manuscript rather explores the general utility of symptomatic Botulinum use in Wilson’s patients but does not address relationships with concurrent copper levels, chelating agent doses, or amount of copper accumulation at the site of action. My questions regarding this case series can be found hereafter:

Major:

  • What relevance the do figures provide in terms of the hypothesis for botulinum dose and copper levels in Wilson’s disease?
  • The role of copper on the Botulinum effectiveness in local and symptomatic use is only relevant if there is a local accumulation of copper that will interact with the toxin. Are there known copper accumulations in the terminal of peripheral nerves? Or in the skeletal muscles? Only few older reports in the literature are noted with no conclusive findings and no copper findings.

Jung KH, Ahn TB, Jeon BS. Wilson disease with an initial manifestation of polyneuropathy. Arch Neurol 2005;62:1628-1631.

Miyakawa T, Murayama E, Sumiyoshi S, Deshimaru M, Miyakawa K. A biopsy case of Wilson's disease. Pathological changes in peripheral nerves. Acta Neuropathol 1973;24:174-177.

The use of DPA is of greater concern as a confounding factor in peripheral neuropathy.

  • What is the relationship between serum copper or 24-hours urine copper and the dose of botulinum in these 6 patients?
  • If higher doses of a chelating agent are used in severe cases, wouldn’t they remove the copper interaction with the Botulinum toxin? Apart from 1-2 non-compliant patients, it could very well be that the DPA controls the copper levels low enough in order not to be of concern.
  • How was the comparison to the literature possible if only one study provided the dose of the botulinum toxin and no studies present copper burden measures?
  • Based on all these questions, I personally do not think there is enough evidence to comment on what Botulinum doses are effective and appropriate and only the general safety/effectiveness of this intervention can be concluded.

Minor:

  • Please use only generic names of drugs throughout the manuscript.
  • Albeit a non-native English speaker myself, there is a need for improving the sentence structure of multiple paragraphs/sentences.

Author Response

Reviewer 1 is right: indeed, we use experimental results (of other authors on the topic of reduction of BoNT action by copper application) as a preface to our case series. However, we do not formulate any hypothesis and do not try to test any hypothesis. No statistics section is included.

Reviewer 1 is right; this is one of the main topics of the manuscript.

In contrast to previous reports on botulinum toxin treatment in WD we present in detail the relevant laboratory findings of all 6 patients in Tab. 1 (including the serum level of copper, the urinary copper excretion) and mention the dose of chelating agents for all 6 patients in the case reports.

The site of action of BoNT is within the cells (nerve cells, gland cells). There is no clinical method available to determine the amount of copper accumulation within cells. Thus, we cannot address this relationship.  

Furthermore, although we describe the largest number of WD-patients treated with BoNT the number of patients is too small to do a reasonable correlation analysis.

The figures demonstrate that our patients were clearly affected. As mentioned above, we present our experience on treatment of symptomatic WD-patients with BoNT.

This topic is not clarified and outside of the present manuscript. We doubt that a local accumulation of copper is necessary to reduce efficacy of BoNT treatment.

Serum copper levels fluctuate from day to day and 24h-urinary copper excretion within a week. Apart from the small number of patients we did not correlate the dose of BoNT with laboratory findings because of these fluctuations.

This is an interesting question. We think and our data suggest that this copper/BoNT interaction does not take place at all in treated WD.

Our patients were non-compliant. Even in these patients BoNT injections were effective.

Indeed, we did not want to make a comparison to the literature but to review the literature.

Reviewer 1 is right: we were also surprised to notice that dose and laboratory findings were not given in detail in previous reports on WD and BoNT.

We think this is a short coming of the previous articles and not of the present manuscript.

We try to do this.

We have tried to improve the language. One of the authors is native speaker.

Reviewer 2 Report

The authors describe 6 patients with diagnosis of Wilson's disease who had botulinum toxin treatment for variety of reasons. 

Overall its interesting to note that there are so few patients reported in literature as WD patients commonly have disabling symptoms due to dystonia, oromandibular dystonia and Sialorrhea. Thus this paper is very important to highlight the importance of Botulinum toxin treatment in symptomatic management of WD. 

The table and figures are excellent!

These are the some suggestions:

1. Case 1 - Recommend to provide further details about the dose of BoNT, duration of response and number of BoNT sessions. Also, kindly comment on degree of improvement. If any clinical scales like Patient global impression of change was done- will be helpful for the readers. Also, did the authors inject for Palmar or axillary or plantar hyperhidrosis. 

2. Case 2 : The history is not clearly presented. “Thereafter she did not take more than 900 mg DPA although she developed a complex segmental dystonia involving the neck and upper trunk muscles during the next years.” It is unclear.  Please provide details of treatment to the patients. 

3. Please provide the details of Botulinum toxin treatment for the other cases as well. 

3. Case 3- did the patient received BoNT for segmental dystonia as well  or only for sialorhea. 

4. Recommend to revise the conclusion: the conclusion of the study is to highlight the importance of BoNT in symptomatic management of WD patients. The present conclusion does not give complete inference of the study. 

Author Response

We totally agree with reviewer 2.

Dose of BoNT is presented in Tab. 1.

More details are presented for this case. 

We inject our patients every 3 months. Therefore we cannot analyse duration of response.

We made a comment on efficacy in the case reports.

We now comment on efficacy in Tab. 1 by adding a line “efficacy”: no effect, mild effect, moderate effect, good effect, excellent effect.

Palmar hyperhidrosis was mentioned in Tab.1.

We have modified and hopefully improved this sentence to:
“During the next years she developed a complex segmental dystonia involving the neck and upper trunk muscles. Nevertheless, she refused to increase the dose of DPA beyond 900 mg”.

This is presented in Tab. 1.

See Part BoNT/A Therapy of Tab. 1.

Reviewer 2 is right: only hypersalivation was treated!

Reviewer 2 is right: we therefore modified the section: Conclusions

Reviewer 3 Report

In this work, the authors provide evidence that despite the inhibitory effect of copper on botulinum neurotoxin A (BoNT/A) seen by using cell or animal-based experiments, in vivo tests on WD's patients did not produce any significantly relevant inhibitory effects on the efficiency of the neurotoxin on such patients. The paper is well written, the data are overall convincing, and the experiments were rigorously carried out.

Author Response

We are pleased about reviewer 3´s summary of the manuscript.

Round 2

Reviewer 1 Report

The Authors did not address any comments in the manuscript. (no track changes are available)

Author Response

Reviewer 1

There is only limited information from pre-clinical studies on how copper might interfere with botulinum toxin.

There is no information on copper levels in the symptomatic but often treated Wilson patients.

There is no information whether copper in Wilson´s patients is expected to be at the presynaptic neuron site of botulinum toxin action.

We have added the following paragraph in the introduction:

A key binding interaction between copper and Cys165 in the BoNT/A LC has been analysed and described. Extracellularly applied ligand-copper complexes at low concentrations effectively reduce intracellular LC-cleavage of SNAP-25. Furthermore, administration of copper complexes in life-threating BoNT/A treated rodents could effectively delay BoNT/A mediated lethality [9].

Both cell-based and animal experiments suggest that efficacy of botulinum toxin treatment may be reduced in patients with Wilson´s disease suffering from deficient copper transport and elevated copper storage.

Information on serum levels of copper and 24h-urinary copper excretion are presented (for our patients) in Tab. 1 under the heading laboratory findings.

We now discuss in detail that the local accumulation of copper is of minor importance compared to the availability of copper intracellularly. By mentioning that the LC is also active in glands we deemphasize the relevance of the presynaptic neuron site.

The reason is not obvious why this is the case. The mechanism how the LC of BoNT/A and copper interact intracellularly is well-known and extracellular administration of copper complexes effectively reduces LC-mediated cleavage of SNAP-25 (9). Thus, for an interaction between copper and the LC of BoNT/A sufficiently high levels of copper have to be available intracellularly. As long as copper is irreversibly bound intracellularly to metallothioneins [10] which is usually the case in long-term treated WD-patients this copper accumulation has no influence on BoNT action. However, the normal response to BoNT/A treatment of muscles and glands in WD as demonstrated in the present paper indicates that the crossly fluctuating and not always elevated levels of free copper in the serum of not optimally cooperating WD-patients as in our series are not high enough to reduce BoNT activity to a clinically relevant extent.

Reply to Editor

While the authors may not have data to answer these questions, they should at least discuss the issues and raise them as possible deficiencies in the paper.

Nobody has clinical data so far to answer these questions.

We pick-up these aspects and discuss them in two new paragraphs: one in the introduction and one in the discussion.

Round 3

Reviewer 1 Report

Thank you.

Author Response

Thank you so much for your comments.